# Biologically Plausible Boltzmann Machine

Arturo Berrones-Santos [1],*,† and Franco Bagnoli [2],†

1. Graduate Program in Systems Engineering and Doctorate in Mathematical Sciences, Autonomous University of Nuevo León, Ciudad Universitaria, San Nicolás de los Garza 66455, Nuevo León, Mexico
2. Department of Physics and Astronomy and CSDC, INFN Sezione di Firenze, University of Florence, Via G. Sansone 1, 50019 Sesto Fiorentino, Italy; franco.bagnoli@unifi.it
* Correspondence: arturo.berronessn@uanl.edu.mx
† These authors contributed equally to this work.

**Abstract:** The dichotomy in power consumption between digital and biological information processing systems is an intriguing open question related at its core with the necessity for a more thorough understanding of the thermodynamics of the logic of computing. To contribute in this regard, we put forward a model that implements the Boltzmann machine (BM) approach to computation through an electric substrate under thermal fluctuations and dissipation. The resulting network has precisely defined statistical properties, which are consistent with the data that are accessible to the BM. It is shown that by the proposed model, it is possible to design neural-inspired logic gates capable of universal Turing computation under similar thermal conditions to those found in biological neural networks and with information processing and storage electric potentials at comparable scales.

**Keywords:** thermodynamic computation; Boltzmann machine; statistical mechanics

## 1. Introduction

The relationship between energy dissipation and information erasure is one of the most firmly established principles in the thermodynamics of computing [1].

In current digital computers, this dissipation is usually orders of magnitude greater than the minimal theoretical limit because of the impossibility of digital logic gates to deal with large thermal fluctuations.

For a digital computer to be capable of correctly storing information at temperature $T$ and to produce an electric signal that overcomes the heat bath noise energy $kT$, where $k$ is the Boltzmann constant, each bit requires electric voltages that depending on the specifics of the chip typically range from ~1 V to ~15 V [2].

Biological information processing systems, on the other hand, which in a very basic sense can be regarded as networks of threshold units connected by electrochemical wires, perform their functions at the mV order of magnitude of the electric voltages and are capable of dealing with the large fluctuations in their living environment.

A training procedure for the Boltzmann machine computational model that naturally leads to network parameters with clear physical interpretation is introduced in this paper.

The general scheme is based on the construction of a prior density of the network's parameters by the maximum entropy principle.

The density is constrained by the observed data encoded like a quadratic Mixed Integer Program (MIP) with a linear continuous substructure for the averages of connections and biases and linear binary substructure for the hidden neurons.

The exploitation of these relatively simple substructures makes the fully connected settings tractable. The possible implications for topics like unsupervised learning, causality and out-of-distribution learning will be outlined and further discussed in a series of forthcoming papers.

In the present contribution, on the other hand, the physical significance of the maximum entropy prior is explored. It is argued that the scheme is not just a mathematical artifact but in fact resembles the way in which actual biological brains perform computations.

It turns out that at least for the basic storage and transmission of bits of information and for the construction of primitive gates capable of universal computation, the resulting physical network and heat bath parameters are consistent with the values observed in biological neural networks. The proposed approach might therefore be a plausible theoretical base on which to eventually build thermodynamic computers [3].

Our article is structured as follows. The required preliminary theoretical concepts, which are the original Boltzmann machine paradigm and the Maximum Entropy Principle, are described in Section 2.1 and Section 2.2, respectively.

Our specific elaboration of a Boltzmann machine with maximum entropy density graph edges (or maximum entropy Boltzmann machine) is given in Section 2.3. Thereafter, the capacity of the maximum entropy Boltzmann machine to store and communicate bits of information among the nodes within the network is demonstrated through the definition of the unit wire in Section 2.4. This is the starting point of our derivation regarding the proposed model as a universal framework for computing. The main results are reported in Section 3.

We start by defining the statistical mechanics of a physical realization of the unit wire in Sections 3.1 and 3.2, and we prove the Turing completeness of our implementation in Section 3.3. Section 3.4 describes a training algorithm that handles fully connected graphs with cycles and therefore with the potential to go beyond Turing machines. The effectiveness of the learning algorithm is tested by the training of a maximum entropy Boltzmann machine for a reversible logic gate with fluctuating connections. A bound for these fluctuations in terms of relevant parameters is provided in Appendix A.

Stochastic computational learning paradigms typically make use of parameters that have no actual physical meaning. Our framework instead makes use of global parameters with direct physical interpretation. The relevance of this for the potential use of the maximum entropy Boltzmann machine as a formal model for biological brains is outlined in Section 4, where additional discussions and conclusions are also given.

## 2. Materials and Methods

### 2.1. The Boltzmann Machine

A Boltzmann machine (BM) is a kind of spin glass [4] with Ising-like variables $x_i \in \{0,1\}$ and external disordered field $h_i$. It is defined on a network whose nodes are labeled by an index $i = 1, \ldots, N$ and with connections $\boldsymbol{q} = \{q_{i,j}\}$.

The energy of a configuration $\boldsymbol{x} = \{x_1, x_2, \ldots\}$ is

$$H(\boldsymbol{x}) = -\left(\sum_{i \neq j=1}^{N} q_{i,j} x_i x_j + \sum_i h_i x_i\right), \tag{1}$$

where the connections are symmetric ($q_{i,j} = q_{j,i}$) to prevent oscillating behaviors.

It is possible to absorb the external field $h_i$ into the self connection $q_{i,i}$ (since for Boolean variables $x_i^2 = x_i$) and thus have

$$H(\boldsymbol{x}) = -\left(\sum_{i,j=1}^{N} q_{i,j} x_i x_j\right). \tag{2}$$

The probability distribution $P(\boldsymbol{x})$ of the system is the Boltzmann one

$$P(\boldsymbol{x}) = \frac{1}{Z} \exp\left(-\frac{H(\boldsymbol{x})}{T}\right), \tag{3}$$

where $T$ is the temperature (in energy units, setting the Boltzmann constant $k = 1$) and

$$Z = \sum_{\boldsymbol{x}} \exp\left(-\frac{H(\boldsymbol{x})}{T}\right) \tag{4}$$

is the partition function (which gives the normalization of the distribution).

The update of the system is implemented by proposing a flip of the the value of a site $i$ and computing the difference of energy

$$\Delta H_i = H(x_0, \ldots, \bar{x}_i, \ldots, x_N) - H(x_0, \ldots, x_i, \ldots, x_N). \tag{5}$$

The flip is accepted the flip with probability

$$p(x_i \to \bar{x}_i) = \frac{\exp\left(-\frac{H(x_0,\ldots,\bar{x}_i,\ldots,x_N)}{T}\right)}{\exp\left(-\frac{H(x_0,\ldots,\bar{x}_i,\ldots,x_N)}{T}\right) + \exp\left(-\frac{H(x_0,\ldots,x_i,\ldots,x_N)}{T}\right)} = \frac{1}{1 + \exp\left(\frac{\Delta H_i}{T}\right)}, \tag{6}$$

which is a logistic-like function.

Random, sequential and parallel implementations of the described general interacting spin dynamics have been proposed, and the further development of algorithms such as Gibbs sampling, parallel tempering or population annealing is an active area of research [5,6]. In any case, the corresponding Markov chain converges to the Boltzmann distribution, since it is ergodic (any state can be reached by any other state) and the flip probability satisfies the detailed balance condition,

$$\exp\left(-\frac{H(x_0, \ldots, x_i, \ldots, x_N)}{T}\right) p(x_i \to \bar{x}_i) = \exp\left(-\frac{H(x_0, \ldots, \bar{x}_i, \ldots, x_N)}{T}\right) p(\bar{x}_i \to x_i). \tag{7}$$

Up to here, this is the description of a disordered magnetic system. From a machine learning standpoint, one defines some units as "visible" and others as "hidden". The system is then trained so that it is able to understand the distribution of the data presented to the visible units (i.e., to extract their correlations and store them into connections) and to generalize the given set based on that distribution, thus originating some generative process of interest [7].

Although the BM model is a very general setup for unsupervised learning, its training in unrestricted fully connected settings is intractable [8]. The origin of this intractability can be traced back to the existence of metastable "glassy" phases if arbitrary non-convex coupling matrices are admitted [9].

In such spin-glass phases, the convergence time to the equilibrium distribution diverges, which effectively breaks down any learning scheme based on Markov Chain Monte Carlo sampling of visible units.

Because of this inherent intractability, only graphs with particular restricted structures had been considered in practice, like for instance directed acyclic networks [8].

The present contribution departs from the traditional distinction between fully unrestricted and restricted models by focusing on the adaptation of the graph structure to the data at hand by the use of the Maximum Entropy Principle.

### 2.2. The Maximum Entropy Principle

Originally introduced to provide a unifying framework for statistical inference, communication theory and thermodynamics [10–12], the Maximum Entropy Principle (MEP) is rooted in Laplace's interpretation of probabilities as degrees of belief or knowledge [13].

Instead of the requirement for frequency data to assign values to probabilities, MEP formalizes the notion that the assignment should consider any prior available evidence and assumptions about the situation which such probabilities are intended to describe, being indifferent to any aspect outside these known evidence and assumptions.

In MEP, the knowledge about the internal states or configurations of a system is

quantified by its entropy. In the absence of any prior knowledge or assumption, the entropy measure results in an equal probability assignment for each configuration. The inclusion of available data or prior knowledge therefore should constrain the entropy.

Our concern in the present work regards the distribution of BM graphs that are consistent with a set of observable data. Given the number of units in the network, each possible graph is characterized by its connectivity matrix, with no prior assumption beside that the couplings between the network's units are real numbers.

As a technical remark, it should be noted that in the present context, the entropy measure is taken over continuous probability spaces. To use the same form of the entropy of its original definition in statistical mechanics and information theory over discrete spaces, a relative entropy is used in this contribution. By taking a uniform density as reference, the resulting expression is the same as that for discrete spaces, simply substituting the summation over configurations with integrals.

*2.3. Maximum Entropy Boltzmann Machine*

Consider a network of $N$ units with binary state space. Each unit depends on all the others by a logistic-type response function. Assume that $d = 1, 2, \ldots, D$ visible binary vectors $x^{(d)} = \{x_i^{(d)}\}$ are presented to the system (we refer collectively to these data with the symbol $\mathcal{D}$). The response of each unit to reproduce a given vector $d$ is defined as

$$\hat{x}_i = 1 \Rightarrow (-1)^{1-x_i^{(d)}} \left[ \sum_j q_{j,i} x_j^{(d)} \right] \geq V_h. \tag{8}$$

$$\hat{x}_i = 0 \Rightarrow (-1)^{1-x_i^{(d)}} \left[ \sum_j q_{j,i} x_j^{(d)} \right] < V_{-h}.$$

The $q$ values are pairwise interactions between units, using the convention for which the self-interactions $q_{i,i}$ are equivalent to shift (or bias) parameters. The voltages $V_h$ and $V_{-h}$ are thresholds associated to the 1 and 0 bit values, respectively.

By introducing the definition

$$Q_{i,d} \equiv (-1)^{1-x_i^{(d)}} \left[ \sum_j q_{j,i} x_j^{(d)} \right], \tag{9}$$

the inequalities (8) are equivalently written as

$$0 < \frac{Q_{i,d}}{\Delta V} \leq 1, \quad i = 0, 1, \ldots, N \quad d = 1, 2, \ldots, D. \tag{10}$$

The denominator in the above expression represents a reference voltage difference, $\Delta V \geq V_h - V_{-h}$. For the network to be capable of reproducing the data $\mathcal{D}$, it should be in a configuration that satisfies Equation (10).

By imposing the condition that the network's topology should comply with Equation (10) on average, there are $g_r = q_r - \langle q_r \rangle = 0$, $r = 1, \ldots, N^2$ constraints to the connectivity parameters. The entropy of the network on the other hand is given by $S = -kT \int P \ln P dP$.

Following [10], the maximum entropy principle under given constraints leads to a probability density for the $q$ values that maximize the Lagrangian,

$$\mathcal{L} = - \int P \ln P d\boldsymbol{q} + \sum_{r=1}^{N^2} \lambda_r g_r(\boldsymbol{q}) + \lambda_0 \left( \int P d\boldsymbol{q} - 1 \right). \tag{11}$$

The first term in the right side of Equation (12) is the entropy relative to a uniform density, the second term accounts for the conditions imposed by the data set through the system of inequalities (10), and the third term represents the normalization condition for the density. The $\lambda$ values are the usual Lagrange multipliers.

Due to the concavity of the entropy, a stationary condition on the Lagrangian applied with respect to $P$ leads to the corresponding maximum entropy probability density,

$$P(\boldsymbol{q}) = e^{\frac{\lambda_0}{kT}} \prod_{r=1}^{N^2} e^{-\frac{\lambda_r}{kT}(q_r - \langle q_r \rangle)}, \tag{12}$$

or, equivalently,

$$P(\boldsymbol{q}) = \prod_{r=1}^{N^2} \alpha_r e^{-\alpha_r(q_r - \beta_r)}, \tag{13}$$

$$\langle q_r \rangle = \frac{1}{\alpha_r} + \beta_r,$$

$$\alpha_r = \frac{\lambda_r}{kT}, \quad \lambda_0 = 0.$$

### 2.4. Implementation of the Unit Wire

The most basic component in both biological and electronic computers is the "unit wire", which permits the preservation in space and time of a unit (or "bit") of information [14].

To develop the formalism in a way that could be plausible for technological implementation or biological realization, the "unit" component stores a bit through a saturation defined by the interval $\Delta V$.

The "wires" $\boldsymbol{q} = (q_{0,0}, q_{0,1}, q_{1,0}, q_{1,1})$, can be seen as edges in a graph like Figure 1, which corresponds to a BM without hidden units under the data set $D = \{(0,0), (1,1)\}$.

The unit wire therefore translates into the MIP,

$$0 < \frac{Q_{i,d}}{\Delta V} \leq 1, \quad i = 0, 1; \quad d = 1, 2. \tag{14}$$

The data set associated to the unit wire leads to

$$0 < [\langle q_{0,1} \rangle + \langle q_{1,1} \rangle] \leq \Delta V \tag{15}$$
$$0 < [-\langle q_{1,1} \rangle] \leq \Delta V$$
$$0 < [\langle q_{1,0} \rangle + \langle q_{0,0} \rangle] \leq \Delta V$$
$$0 < [-\langle q_{0,0} \rangle] \leq \Delta V.$$

In the simplest directed case, $q_{1,0} = 0$. If the bit of information stored at the input $x_0$ is preserved *exactly*, that is, without error, in the output unit $x_1$, it turns out that $Q_{1,D} \equiv \frac{1}{D} \sum_{d=1}^{D} Q_{1,d} = \frac{1}{2} q_{0,1}$. This implies,

$$\left\langle Q_{1,D}^2 \right\rangle - \langle Q_{1,D} \rangle^2 = \frac{1}{4} \left[ \left\langle q_{0,1}^2 \right\rangle - \langle q_{0,1} \rangle^2 \right] \leq \Delta^2 V. \tag{16}$$

The optimal unit wire directed topology therefore should have connectivity parameters taken from a distribution with variance bounded by Equation (16).

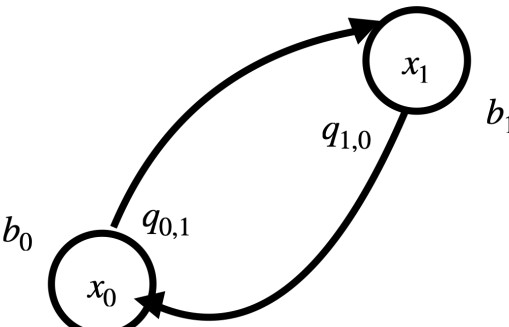

**Figure 1.** Full Boltzmann machine topology for the unit wire. In the figure, $b_0$ and $b_1$ refer to the biases, $q_{0,0}$ and $q_{1,1}$, respectively.

## 3. Results

### 3.1. Statistical Mechanics for a Physical Realization of a Boltzmann Machine

The maximum entropy treatment of the unit wire shows a general feature of the proposed formalism, namely a set of closed relationships between the network's topology, the observed data and the neural activity. More importantly, it shows that these relationships bound the voltages' fluctuations for information storage and processing under dissipation.

Equation (12) or its equivalent form Equation (13), give a self-consistent statistical mechanical description of a Boltzmann machine that is fully connected by electrical wires and is immersed in a heat bath at temperature $T$. The Boltzmann machine and the heat bath conform to a system that interacts with an environment through $N_s \leq N$ sensory units that receive data.

The statistical description of the setup *Boltzmann Machine + Heat Bath + Data* is closed through the solution of the Linear Program (LP),

$$0 < \frac{\langle Q_{i,d} \rangle}{\Delta V} \leq 1, \quad i = 0, 1, \ldots, N_s \quad d = 1, 2, \ldots, D. \tag{17}$$

The LP Equation (17) defines a convex hull in $\mathbb{R}^{N^2 D}$ with the vertices corresponding to the observable units components $\{x_i\}_d; i = 1, 2, \ldots, N_s; d = 1, 2, \ldots, D$.

### 3.2. The Unit Wire Revisited

The LP for the directed unit wire reduces to,

$$0 < \frac{\langle Q_{i,d} \rangle}{\Delta V} \leq 1, \quad i = 0, 1. \quad d = 1, 2. \tag{18}$$

This LP leads to the solution,

$$\langle \boldsymbol{q} \rangle = (\langle q_{0,1} \rangle, \langle q_{1,0} \rangle, \langle q_{0,0} \rangle, \langle q_{1,1} \rangle) = (2\Delta V, 0, 0, -\Delta V). \tag{19}$$

Assuming that the response in electric charge by a unitary change in voltage is the same for all the network's units, the multipliers $\lambda_r$ values can be written $\lambda_r = \lambda = \Delta C$. Under a given temperature and reference voltage for all the neurons and wires, the fluctuations on the network's parameters are therefore governed by

$$P(\boldsymbol{q}) = \frac{1}{Z} e^{-\frac{\Delta C}{kT}(q_{0,1} - 2\Delta V)} e^{-\frac{\Delta C}{kT}(q_{1,1} + \Delta V)}. \tag{20}$$

Equation (20) can be used to completely characterize the neuronal activity. Of particular interest are the fluctuations on the output given an input in the directed unit wire, because these determine the system's ability to preserve and transmit information with zero or small errors. This precision in storage and transmission of information can be characterized by the first two moments of $\hat{x}_1(x_0)$.

In a more general setting, any statistical moment of any particular neuron, group of neurons or function of groups of neurons can be estimated by Monte Carlo sampling of the maximum entropy distribution of the network's parameters.

The first moment of the response of a neuron $x_i$, for instance, is given by,

$$
\langle \hat{x}_i(\{x_{j \neq i}\}) \rangle = \int_q P(\boldsymbol{q}) \left[ \frac{1}{1 + \exp\left(-\frac{\Delta C}{kT} \sum_j q_{j,i} x_j\right)} \right] d\mathbf{q} \tag{21}
$$

$$
\approx \frac{1}{\tau} \sum_{t=1}^{\tau} \text{Sigmoid}_i(\boldsymbol{q}_t).
$$

The numerical implementation of Equation (21) is based on the second form of the maximum entropy density for the connectivity parameters, as shown in Equation (13). If $y$ is a uniform random deviate in the interval $(0, 1)$, any particular parameter $q_r$ can be sampled by the formula,

$$
q_r = -\frac{kT}{\Delta C \cdot 1V} \ln(1 - y) + \beta_r, \tag{22}
$$

$$
\beta_r = \langle q_r \rangle - \frac{kT}{\Delta C \cdot 1V}.
$$

After sampling the entire matrix $\boldsymbol{q}$, a Monte Carlo sampling step is completed by evaluating,

$$
\text{Sigmoid}_i(\boldsymbol{q}) = \frac{1}{1 + \exp\left(-\frac{\Delta C}{kT} \sum_j q_{j,i} x_j\right)}. \tag{23}
$$

The procedure is formally encoded in Algorithm 1.

---

**Algorithm 1** Pseudo-code for Monte Carlo sampling from the maximum entropy distribution.

---

1: Initialize: $\langle q_r \rangle$ from a candidate solution of the MIP feasibility problem obtained analytically or by Algorithm 2.
2: Initialize: $\beta_r = \langle q_r \rangle - \frac{kT}{\Delta C \cdot 1V}$.
3: Assign value to `size` (desired number of samples).
4: Generate $size \times N^2$ uniform and independent random deviates $y$ in the $[0, 1]$ interval.
5: **for** s = 1 **to** size **do**
6:    $q_r = -\frac{kT}{\Delta C \cdot 1V} \ln(1 - y) + \beta_r$
7:    Generate $\boldsymbol{x}_s$ by using **Function**.
8: **end for**
9: **Function**:
10: Pass In: $\boldsymbol{q}$
11: Pass In: $\boldsymbol{x}$
12: Update $\boldsymbol{x}$ by inserting $\boldsymbol{q}$ in Equation (23)
13: Pass Out: $\boldsymbol{x}$
14: **End Function**

---

Numerical experiments have been performed for the directed unit wire considering the in vivo temperature of 310.5 K. The voltage required for a Coulomb of charge to overcome the fluctuations associated with this temperature is equivalent to 26.727 mV.

For the wire voltages, on the other hand, an estimate from the neurophysiology literature of $\Delta V = 70$ mV has been used [15]. With these values,

$$
P(\boldsymbol{q}) = \frac{1}{Z} e^{-\frac{\Delta C}{26.727 \Delta C} (q_{0,1} + q_{1,1} - 70)}. \tag{24}
$$

Figure 2 shows 100 Monte Carlo steps of the unit wire at each of its possible two inputs. It is clear that with the chosen biologically consistent parameters, the unit wire-directed BM responses fluctuate in such a way that the information stored at the input bit is preserved.

Topologies that more closely resemble the kind of structures found in biological neural networks, with full connectivity and the possible inclusion of additional hidden neurons for redundancy, are expected to further improve the signal-to-noise ratio of the storage and transmission of information. The bounds of Equation (A3) discussed in the Appendix A give further theoretical grounds for this assertion. A preliminary algorithm for the training of fully connected models is introduced in Section 3.4. Figure 3 shows the relative difference between the two visible units of a fully connected unit wire topology obtained by the algorithm discussed in Section 3.4. It turns out that as expected, the full connectivity improves the signal-to-noise ratio, which is further enhanced by the inclusion of hidden neurons.

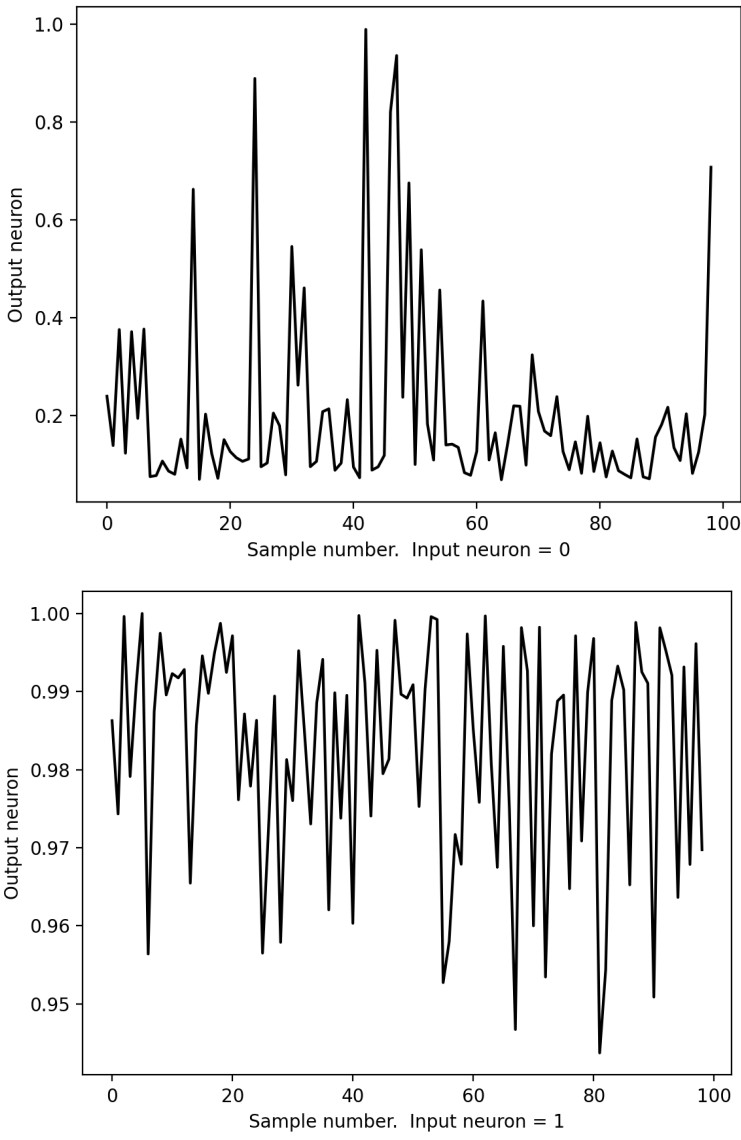

**Figure 2.** Monte Carlo sampling of the output neuron of the directed unit wire.

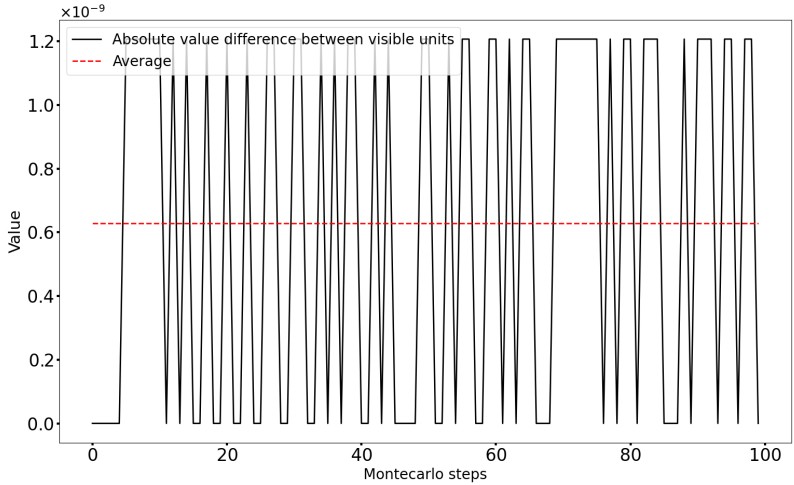

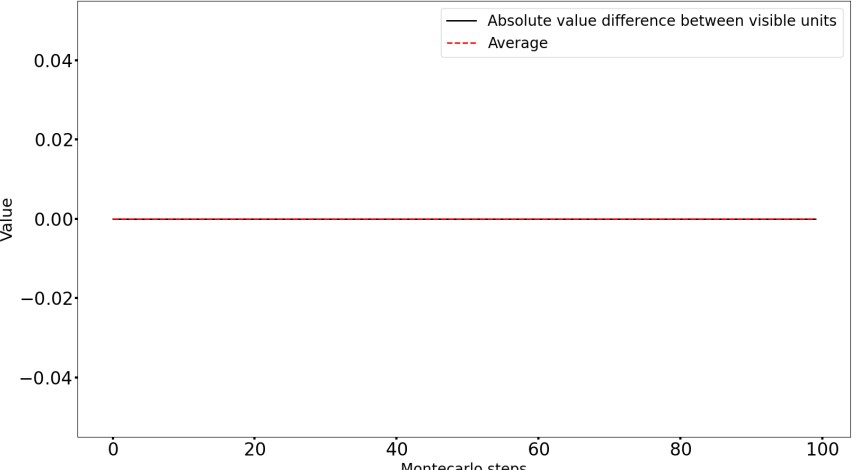

**Figure 3.** Monte Carlo sampling of the visible neurons in fully connected unit wire graphs with no hidden units (**top**) and three hidden units (**botom**).

### 3.3. Turing Completeness

To demonstrate that the proposed formalism contains a Turing complete information processing system, the implementation of the logical negation (NOT, or also named inverter, INV gate), the logical conjunction (AND) and the logical disjunction (OR) is presented. These primitives are thereafter used to construct an exclusive OR (XOR) gate as an example of the universal computation capabilities of the system.

The average outputs and their standard deviations given the inputs have been numerically studied by the use of the prescription given by Formula (21) and its generalization for the estimation of second moments. The code for the numerical experiments can be found in Ref. [16].

The so-called "primitive" gates AND: $(0,0) \rightarrow 0$, $(0,1) \rightarrow 0$, $(1,0) \rightarrow 0$, $(1,1) \rightarrow 1$, OR: $(0,0) \rightarrow 0$. $(0,1) \rightarrow 1$, $(1,0) \rightarrow 1$, $(1,1) \rightarrow 1$, and NOT: $(0) \rightarrow 1$, $(1) \rightarrow 0$, together with the already discussed unit wire ( IDENTITY or UNIT: $(0) \rightarrow 0$, $(1) \rightarrow 1$), are given by directed networks in which the topology parameters have been explicitly constructed by solving the corresponding system of inequalities, as shown in Equation (17).

The directed graph solutions for the primitives are

NOT (INV):

$$\langle q_{0,1} \rangle = -2\Delta V, \tag{25}$$
$$\langle q_{1,1} \rangle = \Delta V,$$
$$\langle q_{i,j} \rangle = 0 \quad \forall \quad (i,j) \neq \{(0,1),(1,1)\}.$$

OR:

$$\langle q_{0,2} \rangle = \Delta V, \tag{26}$$
$$\langle q_{1,2} \rangle = \Delta V,$$
$$\langle q_{2,2} \rangle = -\frac{2}{3}\Delta V,$$
$$\langle q_{i,j} \rangle = 0 \quad \forall \quad (i,j) \neq \{(0,2),(1,2),(2,2)\}.$$

AND:

$$\langle q_{0,2} \rangle = \frac{2}{3}\Delta V, \tag{27}$$
$$\langle q_{1,2} \rangle = \frac{2}{3}\Delta V,$$
$$\langle q_{2,2} \rangle = -\Delta V,$$
$$\langle q_{i,j} \rangle = 0 \quad \forall \quad (i,j) \neq \{(0,2),(1,2),(2,2)\}.$$

These primitives give a Turing-complete information processing system. This is exemplified by the XOR gate: $(0,0) \rightarrow 0$, $(0,1) \rightarrow 1$, $(1,0) \rightarrow 1$, $(1,1) \rightarrow 0$, which is constructed in the following manner: at each Monte Carlo step, a set of network parameters for the AND, OR and INV gates is sampled. The given input $(x_0, x_1)$ is thereafter processed by the Boolean formula

XOR:

$$[x_0 \quad \text{OR} \quad x_1] \quad \text{AND} \quad [\text{NOT} \quad (x_0 \quad \text{AND} \quad x_1)]. \tag{28}$$

Figures 4–11 report the average outputs and their standard deviations given the corresponding inputs for the primitives and the XOR gates. If necessary, the reference potential is modulated in order to have the minimally required voltage separation of one standard deviation to assign the correct output given the input. No modulation is required for the primitives UNIT, INV and OR. The AND primitive displays the minimal distinction requirement at 140 mV, and the XOR composed gate shows the transition to the one standard deviation separation at 200 mV.

### 3.4. Networks with Hidden Neurons and Cyclical Connections for Machine Learning

From the presented results, it might be argued that the reason why the power consumption inefficiency of digital computers is more pronounced in machine learning applications than in other domains is that Turing completeness by irreversible digital gates is not the best suited computation model for learning tasks. Machine learning applications will be discussed in more detail in a series of forthcoming papers.

Here, we simply make a comparison of the XOR gate based on primitives and its machine learning treatment counterpart.

Assume now that the visible units of the Boltzmann machine receive the stream of bit sequences,

$$D = \{(0,0,0),(1,0,1),(0,1,1),(1,1,0)\}. \tag{29}$$

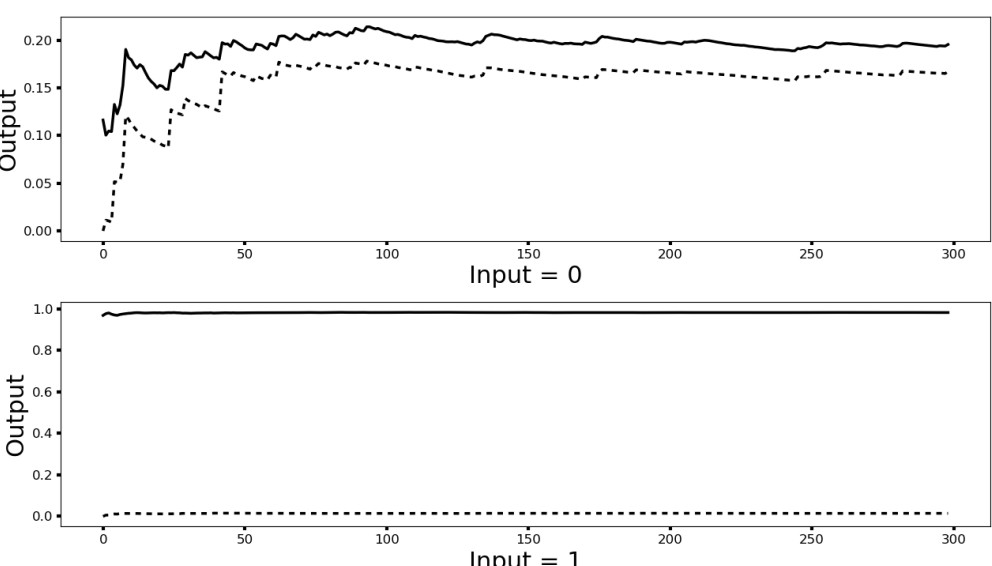

**Figure 4.** Average output (solid line) and standard deviation of the output (dashed line) of the unit wire.

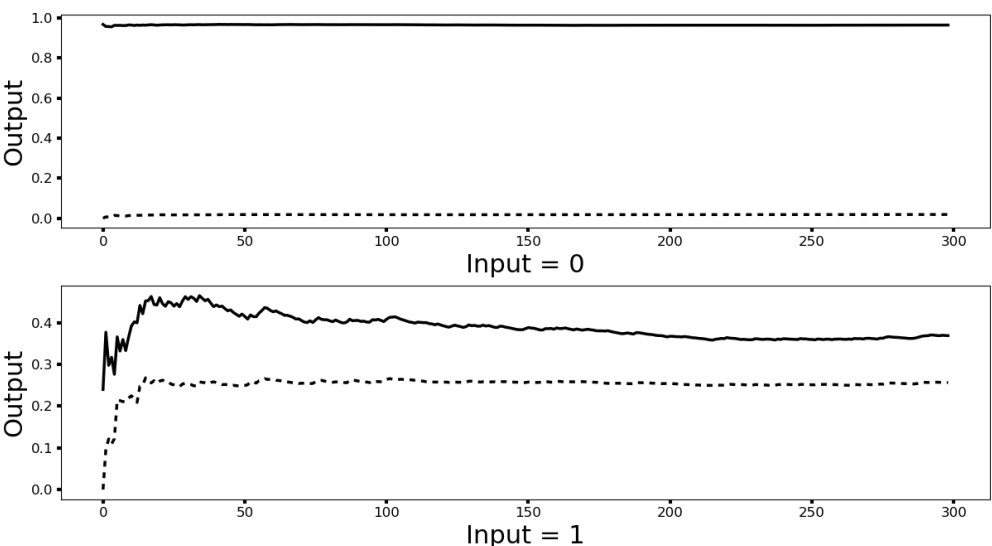

**Figure 5.** Average output (solid line) and standard deviation of the output (dashed line) of the INV gate.

Consider a fully connected Boltzmann machine (BM) with $H$ hidden units. This unrestricted BM setting admits cycles in the associated graph. Therefore, the trained network will give a reversible version of the XOR gate.

To include the presence of hidden units in the Monte Carlo sampling procedure, the following training scheme is used:

1.  Give an initial connectivity matrix $q$ and initial bits for the hidden units for each sequence in $D$.
2.  Generate a candidate connectivity matrix $q_c$ by the use of Equation (22).
3.  Sample new sequences of bits by the application of Sigmoid$_i(q_c)$.

4. If the generated new sequences of bits satisfy a suitable improvement criterion over the visible units, then update $q = q_c$.
5. Repeat until a suitable convergence criterion is met.

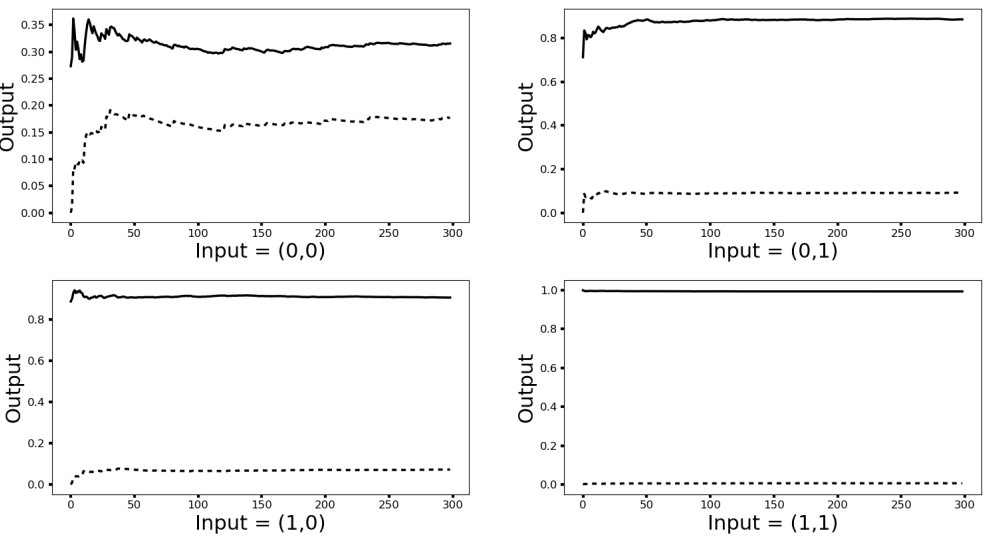

**Figure 6.** Average output (solid line) and standard deviation of the output (dashed line) of the OR gate.

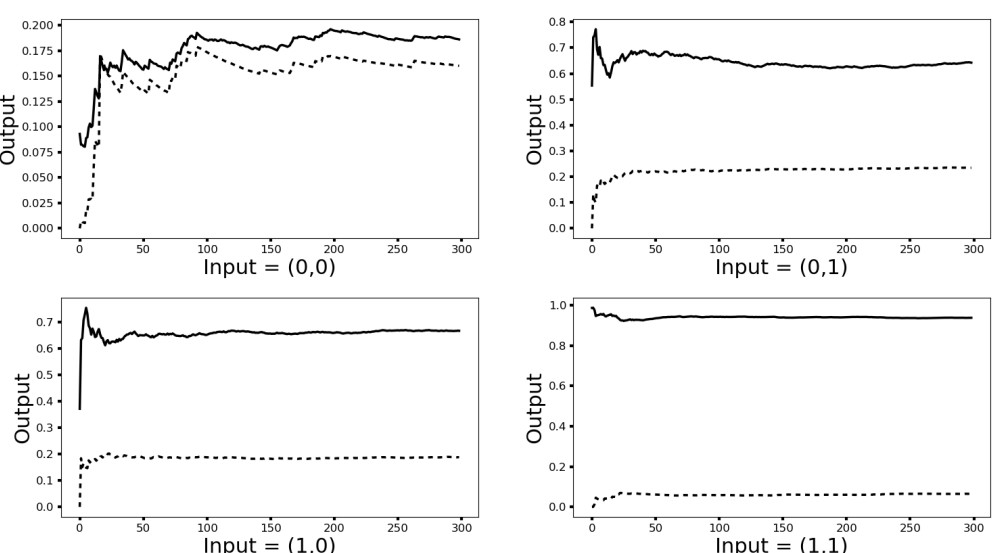

**Figure 7.** Average output (solid line) and standard deviation of the output (dashed line) of the AND gate.

The initial connectivity matrix is proposed through the properties of the LP, as shown in Equation (17). Consider the related set of inequalities,

$$0 \le \frac{Q_{i,d}}{\Delta V} \le 1, \quad i = 0, 1, \ldots, N_s \quad d = 1, 2, \ldots, D. \tag{30}$$

The auxiliary LP given by the set of inequalities Equation (30) has a trivial solution,

namely $q_r = 0 \; \forall \; r$. A probability density for this case should therefore satisfy $\langle q_r \rangle = 0$ and standard deviation equal to $\sigma = \frac{D \Delta V}{\sqrt{N}}$ (see Appendix A). A suitable initial guess for the actual LP Equation (13) can therefore be sampled from a two-parameter exponential density with $\beta_r = 0$ and $\frac{1}{\alpha_r} = \sigma$ for all $r$.

XOR gate, $\Delta V = 70$ mV

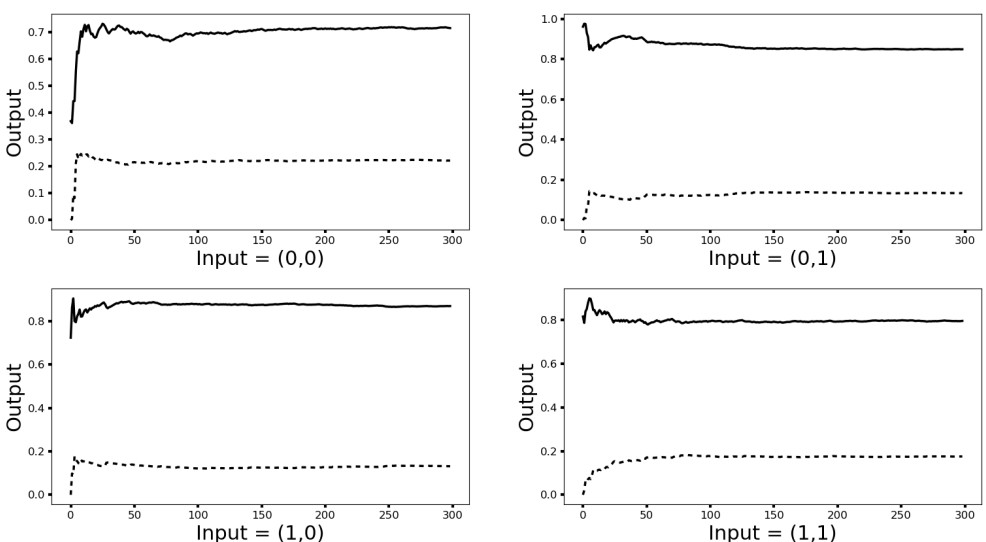

**Figure 8.** Average output (solid line) and standard deviation of the output (dashed line) of the XOR gate at 70 mV.

AND gate, $\Delta V = 140$ mV

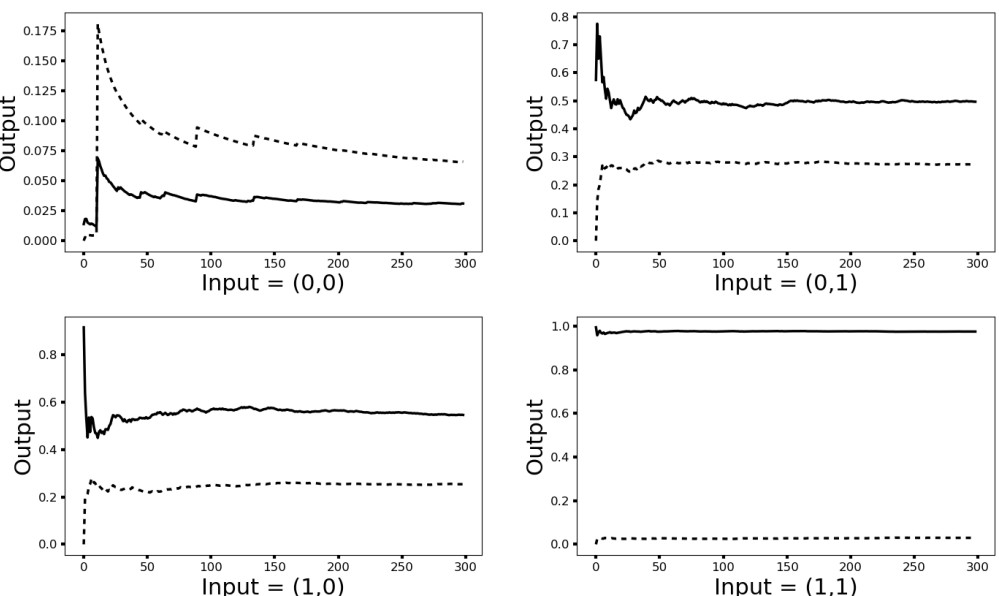

**Figure 9.** Average output (solid line) and standard deviation of the output (dashed line) of the AND gate.

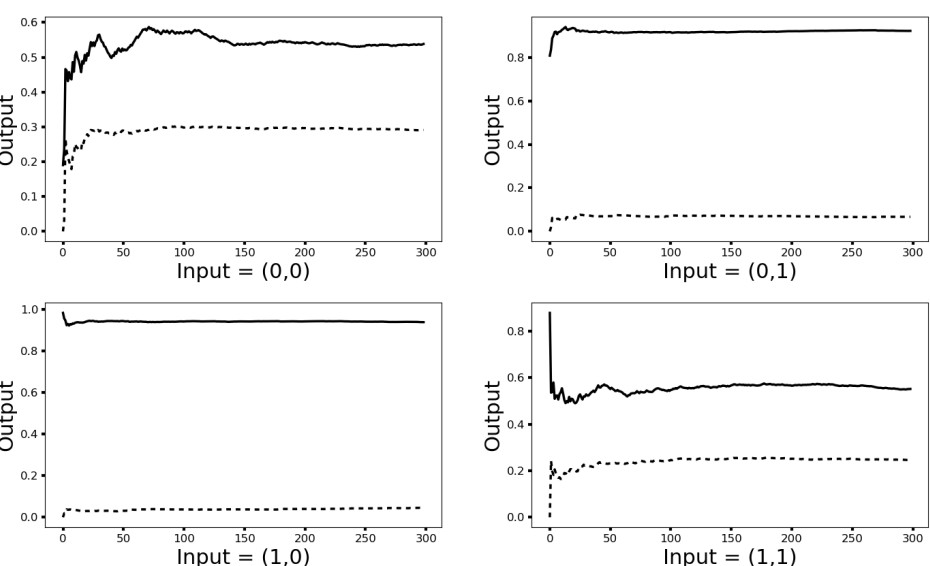

**Figure 10.** Average output (solid line) and standard deviation of the output (dashed line) of the XOR gate.

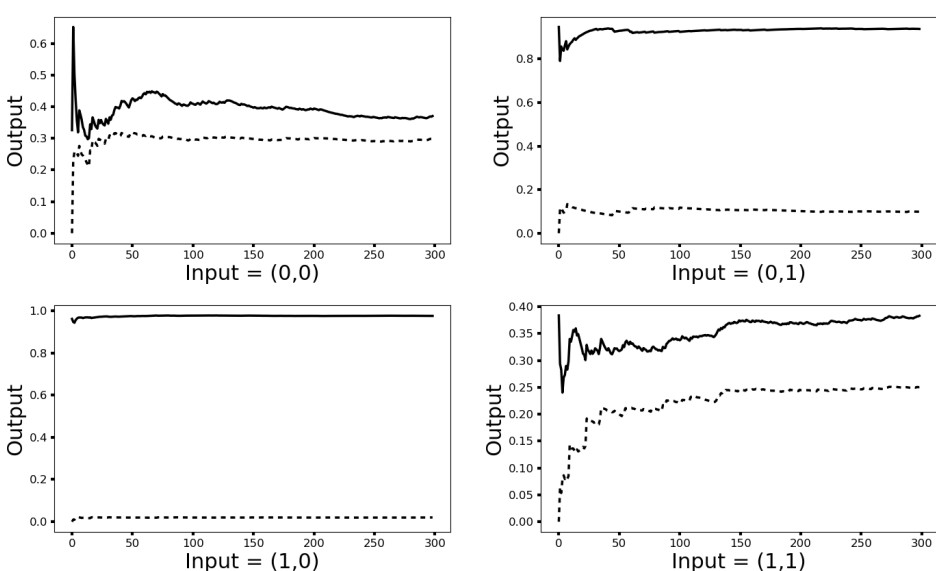

**Figure 11.** Average output (solid line) and standard deviation of the output (dashed line) of the XOR gate.

Once the training is completed, the Monte Carlo sampling is performed as indicated in Section 3.3, using the BM architecture that results from the training procedure. The criteria considered in steps 4 and 5 of the training stage for the XOR experiments are as follows.

Given a candidate graph $q_c$, the first two bits of each sequence in $\mathcal{D}$ are presented to the BM, which generates the remaining (visible and hidden) bits after rounding of the function of Equation (23).

If the difference between the generated and actual third bits in $D$ is reduced, then the candidate graph is accepted and the BM is correspondingly updated.

The training stage is finished when a zero difference between the third visible generated and actual bits in $D$ is obtained.

Monte Carlo sampling is then performed by taking the $q$ that results after training as initial connectivities for the full BM.

The training stage can therefore be viewed as a "thermalization" period for the Monte Carlo sampling. The complete procedure is more thoroughly specified by the pseudo-codes given in Algorithms 1 and 2.

---

**Algorithm 2** Pseudo-code for the thermalization stage of Monte Carlo sampling from the maximum entropy distribution.

1: Initialize: $q_r$ drawn from a two-parameter exponential density with $\beta_r = 0$ and $\frac{1}{\alpha_r} = \frac{D\Delta V}{\sqrt{N}}$. $fitness = 0$. $best\_q = q_r$. $best\_fitness = fitness$. Give a converge threshold for fitness, $conv$.
2: Generate initial hidden variables bits for each $x_d$ in D by drawing from the distribution $p(0) = p(1) = 1/2$.
3: Generate a uniform deviate $y$ in the $[0, 1]$ interval.
4: Update $q_r = -\frac{kT}{\Delta C \cdot 1V} \ln(1 - y) + \beta_r$
5: Generate $\hat{x}_d, d = 1, \ldots, D$, by using **Function**.
6: Sum: summation of the number of different bits between $\hat{x}_d$ and $x_d$. in visible units of interest. If $Sum < 1 - fitness$, then update $fitness = 1 - Sum$. Update $best\_q = q_r$.
7: Repeat until $fitness \geq conv$.
8: **Function**:
9: Pass In: $q$
10: Pass In: $x$
11: Update $x$ by inserting $q$ in Equation (23)
12: Pass Out: $x$
13: **End Function**

---

Figure 12 shows the Monte Carlo sampling given by Algorithms 1 and 2 for a full BM with one hidden unit at a reference voltage of 70 mV.

It is clear that in this reversible setup, the network is capable of a reliable processing of the XOR gates with lesser voltages than its irreversible counterpart.

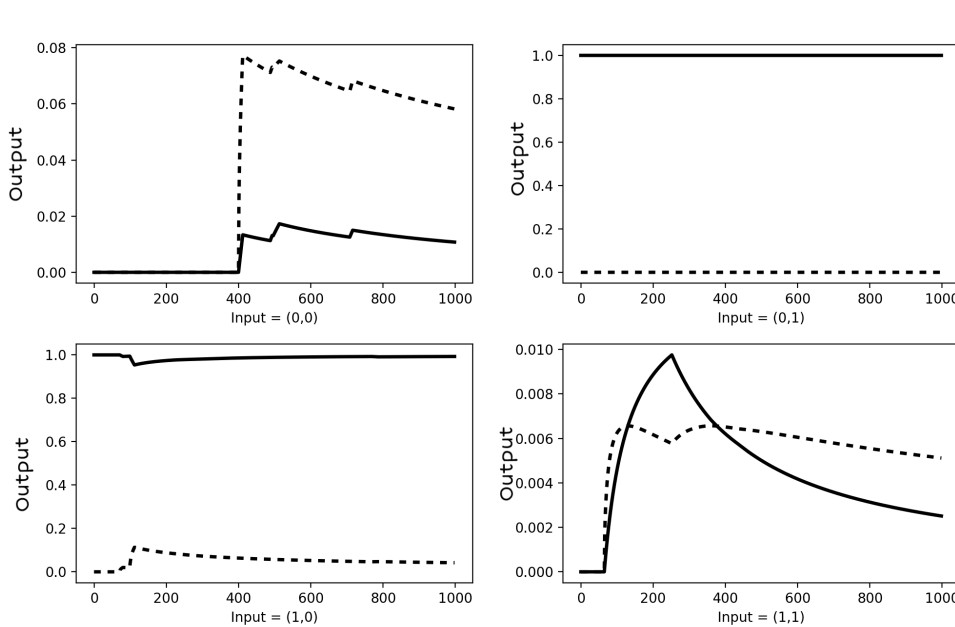

**Figure 12.** Average output (solid line) and standard deviation of the output (dashed line) of the XOR reversible full BM gate at 70 mV.

## 4. Discussion

In this contribution, a model of computation that combines the classical threshold connected units paradigm with statistical mechanics and mathematical programming for data processing is introduced.

The resulting approach leads to a self-consistent statistical description for the voltages that characterize the physical connections of a Boltzmann machine under a given heat bath and data.

These voltages can be modulated in regard to the information processing task at hand in a well-defined manner.

The given formalism leads to the possibility of universal Turing computation by Boltzmann machines with directed and also reversible topology at room or biologically relevant temperatures. The resulting distribution of couplings that characterize the Boltzmann machine implies electric voltages differences comparable to those found in biological neural networks.

The presented results might indicate a plausible route for the construction of thermodynamic computers implemented on ad hoc electronic circuits or with emergent neuromorphic hardware such as spintronic chips [17] or memristors [18].

From a more biological standpoint, the picture that emerges is that the proposed maximum entropy Boltzmann machine (MEBM) represents a formal model for computation that agrees with two key parameters that characterize the mammalian brain, namely the temperature at which computations are performed and (after changing the conventional sign of the reference voltage in inequalities Equation (17)) the resting potential of the computational units.

A more subtle biological feature also captured by our model is that in contrast to previous Boltzmann machine frameworks in which the disorder of the graph couplings is "quenched", the couplings of the MEBM fluctuate in such a way that give statistically reliable neuron responses. This is reminiscent of the correlations' variability inferred from time series of firing neurons. For instance, according to [19]:

"The most straightforward interpretation of the widely observed relationship between the activity of neurons in sensory cortex and animals' behavioral choices is that random fluctuations in the activity of sensory neurons influence perceptual decisions. If the decision is supported by large pools of neurons (more than ∼100 neurons), these random fluctuations must be correlated between members of a pool. Understanding which signals give rise to these correlations is therefore central to the interpretation of choice probabilities. The results of multiple studies suggest that the correlation structure is not fixed but depends on the task an animal performs. Recent evidence suggests that, at least for some tasks, part of this signal reflects the influence of cognitive factors on sensory neurons, but there is currently no agreed upon method that allows the relative magnitude of flexible top-down and hard-wired bottom-up components to be quantified".

In our model, the statistical properties of the couplings between neurons depend on the task provided to the BM. These statistical properties are completely characterized in terms of a probability density that, besides the task, depends on the already mentioned observable macroscopic quantities of the neuronal network, the in vivo temperature and the global reference voltage. An important avenue for further research will therefore be the study of the variability of the neurons' correlations within our framework in large-scale BMs under data streams related to tasks of cognitive interest.

A central open question in machine learning is the effective handling of unlabeled data. The construction of balanced representative data sets for supervised machine learning for the most part still requires a very close and time-consuming human direction. Other related currently pressing machine learning problems are causal relationships discovery and out-of-distribution generalization. These questions are all related to unsupervised tasks. Because the full Boltzmann machine paradigm provides a general framework to perform unsupervised learning, another future research question valuable from a purely

computational perspective is the development of efficient learning from data algorithms in an unsupervised fashion.

**Author Contributions:** Conceptualization, A.B.-S. and F.B.; Methodology, A.B.-S. and F.B.; Software, A.B.-S.; Validation, A.B.-S. and F.B.; Writing—Original Draft Preparation, A.B.-S.; Writing—Review and Editing, F.B. All authors have read and agreed to the published version of the manuscript.

**Funding:** A.B.-S. acknowledges partial financial support by UANL-PAICYT under grant IT1807-21 and by CONACYT under grant CB-167651.

**Institutional Review Board Statement:** Not applicable.

**Informed Consent Statement:** Not applicable.

**Data Availability Statement:** The software that generates the numerical experiments presented in the paper can be downloaded from https://github.com/ArturoBerronesSantos/bioplausBM (accessed on 30 March 2023).

**Conflicts of Interest:** The authors declare no conflict of interest.

## Appendix A. Bound for the Couplings Fluctuations

In this technical appendix, the bounded domain used for the initialization of the learning algorithm introduced in the machine learning treatment of the reversible XOR gate, Section 3.4, is justified. Moreover, it is argued that the derived bound in the fluctuations in the couplings parameters of the graph nodes will be relevant for the generalization of the learning procedure to large-scale networks.

The starting point is the mixed-integer program, as shown in Equation (10). Consider the definition,

$$Q_{1,D} \equiv \frac{1}{D} \sum_{d=1}^{D} Q_{1,d}. \tag{A1}$$

By writing the mixed-integer program Equation (10) in terms of the quantity defined by Equation (A1), it follows that

$$\sum_{d}^{D} \sum_{j}^{N} q_{i,j} x_j \leq D \Delta V. \tag{A2}$$

In Section 2.3, it has been shown that the maximum entropy distribution that characterizes the network's topology is written as a product of independent densities for each node coupling if there exist solutions for set of inequalities Equation (10) under the data set. Combining this with the observation that the maximum possible variance in the couplings is attained in a configuration with all the network's active nodes, the following bound for the variance in the couplings' fluctuations is obtained,

$$\left\langle q_{i,j}^2 \right\rangle - \left\langle q_{i,j} \right\rangle^2 \leq \frac{(D \Delta V)^2}{N}. \tag{A3}$$

The expression given by Equation (A3) relates the variability in the internal structure of the network with the observed data, the network's size and the global reference voltage. From the maximum entropy density given by Equations (12) and (13), the expression Equation (A3) is equivalent to

$$\left( \frac{kT}{\lambda_{i,j}} \right)^2 \leq \frac{(D \Delta V)^2}{N} \tag{A4}$$

In the above expression, the multiplier $\lambda_{i,j}$ can be interpreted like the electrical con-

ductivity between nodes *i* and *j* by using the typical duration of an interaction between the nodes as the time unit.

In addition to the application of the bound Equation (A3) for the initialization of Algorithm 2 presented in Section 3.4, it might be expected to be relevant to issues regarding the maximum entropy Boltzmann machine on large-scale networks and data sets. For instance, the statistical independence between couplings arises as a consequence of the satisfaction of the constraints imposed by the data. The bounds Equations (A3) and (A4) therefore provide a condition for the topological and physical quantities that characterize the network in order for such a network to be able to learn and generalize from the given data, which might then result in a useful guide to assess the convergence and improve the efficiency of the training of the MEBM in large-scale situations. A particular implementation of these ideas that is intended to be developed in future works is by the construction of an evolution equation for the graph's connectivities $q$. The bound Equation (A3) defines a maximum entropy stationary state that by the inequalities (10) is linked to the learnability of the data set. The learning process defined in this way is an evolution through correlated graph configurations that tend to a stationary state, which corresponds to the maximum entropy description of the data. This approach would also permit relating the MEBM formalism to generalized evolutionary models based on entropy production like those discussed in [20,21].

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
