# Peer review of "Biologically Plausible Boltzmann Machine"

_informatics, doi:10.3390/informatics10030062_

Round 1
Reviewer 1 Report
The authors develop ideas that may lead to a Boltzmann Machine (BM) capable of computation at ambient temperatures and low energies. Their ideas seem quite novel and plausible, and the preliminary results are promising. I recommend publication of the manuscript in Informatics after the authors have considered the following questions.
Pg. 3, just above Eq. (5) it is stated that the sites may be chosen randomly, or sequentially. Sequential choice of sites often yields large fluctuations and/or long correlations. Have the authors observed this? Which choice might be preferred during a simulation?
Pg. 9, Fig. (4). The input should be inverted for the INV gate. The I=0 input is well-characterized by O=1, with a small standard deviation, whereas I=1 yields 0~0.4, with a large standard deviation. Similarly, large uncertainties occur in most other gates. The authors might comment about how such variations might be accommodated in a practical application.
Pg. 14, line 300. The authors mention machines may have “directed and also reversible topology.” This sounds interesting. Could the authors expand on the ingredients needed to make the machine reversible? The MC process they describe is usually irreversible.
Pg. 15, line 333. I strongly agree with the statement that “An important avenue for further research will therefore be the study of the neuron correlations variability…” It is know that variable interactions can greatly increase the entropy, so I am curious about what is known and how easy it might be to implement variable correlations in the authors’ model.
Minor grammar issues should be corrected before publication.
Author Response
*Remark 1
> The authors develop ideas that may lead to a Boltzmann
> Machine (BM) capable of computation at ambient
> temperatures and low energies.
> Their ideas seem quite novel and plausible, and the
> preliminary results are promising.
> I recommend publication of the manuscript in
> Informatics after the authors have considered the
> following questions.
> Pg. 3, just above Eq. (5) it is stated that the sites may be
> chosen randomly, or sequentially.
> Sequential choice of sites often yields large fluctuations
> and/or long correlations.
> Have the authors observed this? Which choice might be
> preferred during a simulation?
*Answer:
An additional paragraph has been added in the corresponding section (below Eq. (6)),
with relevant pointers to the literature on different algorithmic implementation of the updating rules and simulation of magnetic systems.
For the small dimensional situations considered in the article the details on updating rules appear to be not so pressing but the discussion regarding the scaling of the approach presented in the Appendix has been expanded
and more details on the intended developments for large dimensional problems are now included.
*Remark 2:
> Pg. 9, Fig. (4). The input should be inverted for the INV
> gate. The I=0 input is well-characterized by O=1,
> with a small standard deviation, whereas I=1 yields
> 0~0.4, with a large standard deviation.
> Similarly, large uncertainties occur in most other gates.
> The authors might comment about how such variations
> might be accommodated in a practical application.
*Answer:
Additional experimentation that address this important question is given at the end of section 3.2. It's shown that the signal to noise ratio is improved by considering fully connected topologies and by the inclusion of hidden neurons. This assertion is theoretically justified
by the results presented in the Appendix section, that is now expanded by giving more details on the possible algorithmic routes for the scaling to large dimensional graphs.
*Remark 3:
> Pg. 14, line 300. The authors mention machines may
> have “directed and also reversible topology.”
> This sounds interesting.
> Could the authors expand on the ingredients needed to
> make the machine reversible?
> The MC process they describe is usually irreversible.
*Answer:
The algorithm for the training of cyclic graphs given in
Section 3.4 is further tested on networks with hidden units and its effectiveness for the improvement of the signal to noise ratio in the visible neurons of interest is experimentally presented in the above mentioned new results of Section 3.2.
*Remark 4:
> Pg. 15, line 333. I strongly agree with the statement that
> “An important avenue for further research will therefore
> be the study of the neuron correlations variability…”
> It is know that variable interactions can greatly increase
> the entropy, so I am curious about what is known and
> how easy it might be to implement variable correlations
> in the authors’ model.
*Answer:
This issue is now more thoroughly discussed, at least from an algorithmic standpoint, in the Appendix, where a learning scheme based on an evolution equation of the probability density of the graph's interaction matrix is proposed. In the stationary state of this evolution the entropy is maximized. This is intended to be the starting
point of our subsequent work which involves large scale problems.
* Remark 5:
> Minor grammar issues should be corrected before
> publication.
* Answer:
We agree, thank you; this has been done.
Reviewer 2 Report
This paper presents a study of a type of Boltzmann machine that has been designed to implement Turing universal computation through the use of a maximum entropy assumption for defining the priors on the system's parameters.
Overall, this is an interesting contribution to the learning and computation literature, and I am interested to see the forthcoming papers that the authors mentioned.
From some of the figures 3-10, it would appear that some of the logical operations are difficult to achieve (outputs a little far from their targets or standard deviation spreading the possible output into the wrong region (e.g., too low). Perhaps this could be clarified in the text.
Perhaps normalize all y axes of fig’s 3-10 to [0,1] to allow easier comparison between the curves of these figures. If you need additional figures to show finer details, maybe use extra figures or inset figures.
In section 2.4, perhaps mention the connection to rectification here.
Please also see minor suggestions in the attached pdf.

I recommend a final sweep for typos or minor errors.
Author Response
* Remark 1:
> This paper presents a study of a type of Boltzmann
> machine that has been designed to implement Turing
> universal computation through the use of a maximum
> entropy assumption for defining the priors on the
> system's parameters.
> Overall, this is an interesting contribution to the learning
> and computation literature,
> and I am interested to see the forthcoming papers that
> the authors mentioned.
> From some of the figures 3-10, it would appear that
> some of the logical operations are difficult to achieve
> (outputs a little far from their targets or standard
> deviation spreading the possible output into the wrong
> region (e.g., too low). Perhaps this could be clarified in
> the text.
> Perhaps normalize all y axes of fig’s 3-10 to [0,1] to allow
> easier comparison between the curves of these figures.
> If you need additional figures to show finer details,
> maybe use extra figures or inset figures.
* Answer:
Additional results at the end of section 3.2. are presented which address this concern.
It's shown that the signal to noise ratio is improved by considering fully connected topologies and by the inclusion of hidden neurons. This assertion is theoretically justified by the results presented in the Appendix section, that is now expanded by giving more details on the possible algorithmic routes for the scaling to large dimensional graphs.
*Remark 2:
> Please also see minor suggestions in the attached pdf.
*Answer:
Thank you; this has been done.
*Remark 3:
> I recommend a final sweep for typos or minor errors.
*Answer:
We agree, thank you; this has been done.
Reviewer 3 Report
The aim of this paper is to provide a Boltzmann Machine approach to computation trough an electric substrate under thermal fluctuations and dissipation.
The paper is very interesting. I suggest only some improvements:
- Eq. (1) q_{i,j} but in Eq. (2) q_{ij}: please uniform the symbols
- Eq. (11): please provide a Reference or derive the relations
In relation to the statistical approach I consider useful to introduce some results from the following papers:
- Bejan, A. Discipline in Thermodynamics. Energies 2020, 13, 2487. Doi: 10.3390/en13102487
- Miguel, A. F.; Bejan, A. The principle that generates dissimilar patterns inside aggregates of organisms. Physica A 2009, 388, 727. Doi: 10.1016/j.physa.2008.11.013
- Lucia, U. Thermodynamic paths and stochastic order in open systems. Physica A 2013, 392, 3912. Doi: 10.1016/j.physa.2013.04.053
- Grisolia, G.; Lucia, U. Why does thermomagnetic resonance affect cancer growth? A non‑equilibrium thermophysical approach. Journal of Thermal Analysis and Calorimetry 2022, 147, 5525. Doi: 10.1007/s10973-022-11294-8
- Lucia, U. The Gouy-Stodola theorem as a variational principle for open systems. Atti dell'Accademia Peloritana dei Pericolanti di Messina 2016, 94, A4. Doi: 10.1478/AAPP.941A4
- Wada T, Scarfone AM. On the Kaniadakis Distributions Applied in Statistical Physics and Natural Sciences. Entropy. 2023; 25(2):292. Doi: 10.3390/e25020292
- Kaniadakis, G. Maximum entropy principle and power-law tailed distributions. Eur. Phys. J. B 2009, 70, 3. Doi: 10.1140/epjb/e2009-00161-0
Author Response
*Remark 1:
> The aim of this paper is to provide a Boltzmann Machine
> approach to computation trough an electric substrate
> under thermal fluctuations and dissipation.
> The paper is very interesting. I suggest only some
> improvements:
> - Eq. (1) $q_{i,j}$ but in Eq. (2) $q_{ij}$ : please uniform
> the symbols
* Answer:
We agree, thank you; this has been done.
*Remark 2:
> - Eq. (11): please provide a Reference or derive the
> relations
*Answer:
A new paragraph below Eq. (11) which clarifies its derivation is now included.
*Remark 3:
> relation to the statistical approach I consider useful to
> introduce some results from the following papers:
> - Bejan, A. Discipline in Thermodynamics. Energies
> 2020, 13, 2487. Doi: 10.3390/en13102487
> - Miguel, A. F.; Bejan, A. The principle that generates
> dissimilar patterns inside aggregates of organisms.
> Physica A 2009, 388, 727. Doi:
> 10.1016/j.physa.2008.11.013
> - Lucia, U. Thermodynamic paths and stochastic order in
> open systems. Physica A 2013, 392, 3912. Doi:
> 10.1016/j.physa.2013.04.053
> - Grisolia, G.; Lucia, U. Why does thermomagnetic
> resonance affect cancer growth?
> A non‑equilibrium thermophysical approach. Journal of
> Thermal Analysis and Calorimetry 2022, 147, 5525. Doi:
> 10.1007/s10973-022-11294-8
> - Lucia, U. The Gouy-Stodola theorem as a variational
> principle for open systems.
> Atti dell'Accademia Peloritana dei Pericolanti di Messina
> 2016, 94, A4. Doi: 10.1478/AAPP.941A4
> - Wada T, Scarfone AM. On the Kaniadakis Distributions
> Applied in Statistical Physics and Natural Sciences.
> Entropy. 2023; 25(2):292. Doi: 10.3390/e25020292
> - Kaniadakis, G. Maximum entropy principle and power-
> law tailed distributions. Eur. Phys. J. B 2009, 70, 3. Doi:
> 10.1140/epjb/e2009-00161-0
*Answer:
Thank you for this very valuable suggestion. Some of the recommended references have been linked to the discussion in the updated Appendix section.